# Cyclodextrin Diethyldithiocarbamate Copper II Inclusion Complexes: A Promising Chemotherapeutic Delivery System against Chemoresistant Triple Negative Breast Cancer Cell Lines

**DOI:** 10.3390/pharmaceutics13010084

**Published:** 2021-01-10

**Authors:** Ammar Said Suliman, Mouhamad Khoder, Ibrahim Tolaymat, Matt Webster, Raid G. Alany, Weiguang Wang, Abdelbary Elhissi, Mohammad Najlah

**Affiliations:** 1Pharmaceutical Research Group, School of Allied Health, Faculty of Health, Education, Medicine and Social Care, Anglia Ruskin University, Bishops Hall Lane, Chelmsford CM1 1SQ, UK; ammar.said-suliman@aru.ac.uk (A.S.S.); ibrahim.tolaymat@aru.ac.uk (I.T.); matt.webster@aru.ac.uk (M.W.); 2School of Life Sciences, Pharmacy and Chemistry, Kingston University London, Kingston Upon Thames, London KT1 2EE, UK; m.khoder@kingston.ac.uk (M.K.); r.alany@kingston.ac.uk (R.G.A.); 3Faculty of Science & Engineering, University of Wolverhampton, Wolverhampton WV1 1LY, UK; w.wang2@wlv.ac.uk; 4College of Pharmacy and Office of the Vice President for Research and Graduate Studies, Qatar University, P.O. Box 2713, Doha, Qatar; aelhissi@qu.edu.qa

**Keywords:** diethyldithiocarbamate copper II, beta-cyclodextrin, chemoresistance, triple negative breast cancer, solubility

## Abstract

Diethyldithiocarbamate Copper II (DDC-Cu) has shown potent anticancer activity against a wide range of cancer cells, but further investigations are hindered by its practical insolubility in water. In this study, inclusion complexes of DDC-Cu with hydroxypropyl beta-cyclodextrin (HP) or sulfobutyl ether beta-cyclodextrin (SBE) were prepared and investigated as an approach to enhance the apparent solubility of DDC-Cu. Formulations were prepared by simple mixing of DDC-Cu with both cyclodextrin (CDs) at room temperature. Phase solubility assessments of the resulting solutions were performed. DDC-Cu CD solutions were freeze-dried for further characterisations by DSC, thermogravimetric analysis (TGA) and FT-IR. Stability and cytotoxicity studies were also performed to investigate the maintenance of DDC-Cu anticancer activity. The phase solubility profile deviated positively from the linearity (Ap type) showing significant solubility enhancement of the DDC-Cu in both CD solutions (approximately 4 mg/mL at 20% *w/w* CD solutions). The DSC and TGA analysis confirmed the solid solution status of DDC-Cu in CD. The resulting solutions of DDC-Cu were stable for 28 days and conveyed the anticancer activity of DDC-Cu on chemoresistant triple negative breast cancer cell lines, with IC_50_ values less than 200 nM. Overall, cyclodextrin DDC-Cu complexes offer a great potential for anticancer applications, as evidenced by their very positive effects against chemoresistant triple negative breast cancer cells.

## 1. Introduction

Diethyldithiocarbamate copper (II) (DDC-Cu) consists of two molecules of diethyldithiocarbamic acid (DDC), a disulfiram (DS) metabolite, with one ion of copper II [1,2]. DS, clinically used as an anti-alcoholism drug, has been recently shown to possess potent anticancer activities against a wide range of cancers including colon, breast, lung, prostate, ovarian, cervical and brain cancers [1,3,4]. Additionally, the anticancer activity of DS includes effective termination of drug-resistant cancer stem cells (CSCs) and reverses chemoresistance [3,5]. However, the biological instability of DS in bloodstream has severely limited its clinical use in anti-cancer therapy [3]. Disulfiram chelates copper II to produce the complex DDC-Cu (Figure 1). The formation of this complex generates reactive oxygen species (ROS), inducing cancer cell death [1]. However, the anticancer activity of DSF/Cu is not limited to instantaneous stress induced by ROS but also includes other mechanisms such as the inhibition of NF-κB pathways [2,6]. The anti-cancer activity of DSF is completely dependent on the supplementation of copper II [2,7]. Therefore, the preformed complex DDC-Cu itself has shown strong cytotoxicity against cancer cells when applied directly [1]. Interestingly, it has been reported that DS-copper complex (DDC-Cu) is a potent proteasome inhibitor inducing apoptosis in cancer cells but not normal, immortalized breast epithelial cells [2,8]. More recently, Liu et al. have shown that DS/Cu is specifically toxic to breast cancer cell lines (MCF-7 and T47D) and spares normal cell lines (HeCV and MCF10A) [9].

However, the realisation of DDC-Cu as a new anticancer therapeutic agent is limited by its very poor solubility in water. Previous attempts to circumvent this issue are mainly based on a “split” delivery of both DDC (water-soluble) and a solution of copper salt. The reaction between both was meant to take place at the cancer cells to produce the anti-cancer action of DDC-Cu [10]. However, this reaction was not localised to cancer tissue and insoluble crystals of DDC-Cu were still instantly formed, reducing the effective fraction [1]. Recently, Marengo et al. have followed similar “split” approach but by loading copper salts to the lipid phase of PEGylated hyaluronic acid liposome encapsulating saturated solution of DDC-sodium [11]. This method might theoretically offer a slight enhancement in the solubility of DDC-Cu in liposomal dispersions; however, no long-term stability was proven. The lipid bilayer would be depleted of Cu^2+^ chelated by DDC, consequently, DDC-Cu crystals might be precipitated or adsorbed onto the external liposomal surfaces [11].

CDs are cyclic oligosaccharides produced by the enzymatic degradation of starch, where glucose residues link together by α-1,4 glycosidic bonds and form macrocyclic molecules. According to the number of glucose subunits, CDs are categorized into α-CD, β-CD, and γ-CD, consisting of 6, 7, and 8 glucopyranose units, respectively [12,13]. The unique truncated-cone-shaped structure of CDs, with hydrophilic surfaces and hydrophobic cavities, enabled them to form inclusion complex with poorly soluble drugs. The resulting CD-drug inclusion complexes are known to enhance drug solubility, aqueous stability, and shelf-life [14,15,16]. Furthermore, hydroxypropyl beta-cyclodextrin (HP) and sulfobutyl ether beta-cyclodextrin (SBE) were classified by FDA as inactive pharmaceutical excipients, i.e., both could be used in parenteral formulations with lower cytotoxic effects compared to that of β-CD [14,17,18].

Despite the predominant interest in the wide spectrum anticancer activity of disulfiram in the presence of copper, there has been limited research on proposing the resulting complex DDC-Cu for cancer treatment. This is mainly due to the fact that this anticancer material is not only practically insoluble in water but also cumbersome when loaded to liquid formulations. Therefore, the development of stable water solution of DDC-Cu is needed to allow further investigations on the potential use of this complex in the treatment of cancer. In this study, we aim to report the preparation and characterization of inclusion complexes formed by DDC-Cu and beta-CD derivatives HP or SBE for enhanced apparent solubility of DDC-Cu. The resulting formulations were freeze-dried (lyophilized) to assess drug-in-solution stability and amenability to be used in novel anticancer applications. The impact of inclusion on the cancer activity were also evaluated by MTT assay using triple negative breast cancer cell lines.

## 2. Materials and Methods

### 2.1. Materials

Copper (II) Diethyldithiocarbamate (DDC-Cu) (molecular weight = 360.07 g/mole and +97.0% purity) was purchased from Tokyo Chemical Industry Co., Ltd., Tokyo, Japan. HP (molecular weight = 1555 g/mol) and SBE (molecular weight = 2242.05 g/mole) were both purchased from Glentham, Wiltshire, UK. M.D. Anderson-Metastasis Breast cancer (MDA-MB-231) and MDA-MB-231 resistant to paclitaxel (PAC) at 10 nM concentration (MDA-MB-231_PAC10_) were obtained from Professor Weiguang Wang’s research Group, University of Wolverhampton, Wolverhampton, UK. In detail, MDA-MB-231_PAC10_ was developed from MDA-MB-231 (purchased from ATCC, Middlesex, Teddington, UK) using a procedure reported previously [5]. Paclitaxel (+99.5%) was obtained from Alfa Aesar, Lancashire, UK. Dulbecco’s modified Eagle’s medium (DMEM), Gibco™ fetal bovine serum (FBS), penicillin streptomycin (Pen-Strep) antibiotic solution, non-essential amino acid solution and L-glutamine (cell culture tested, 99.0–101.0%), ethanol (absolute and 70%), 50 mL centrifuge tubes (sterile), tissue culture flask 75 cm^2^ (sterile) and serological pipettes (sterile) Gibco™ Trypsin-EDTA (0.25%), phenol red, sodium chloride (99.5%), potassium chloride, potassium phosphate and sodium phosphate were purchased from Fisher Scientific, Loughborough, UK. 3-(4,5-Dimethylthiazol-2-Yl)-2,5-Diphenyltetrazolium Bromide (MTT) and Dimethyl sulfoxide (DMSO) were purchased form Glentham Life Science, Wiltshire, UK. Corning^®^ 96 Well TC-treated microplates were obtained from Sigma-Aldrich, Dorset, UK.

### 2.2. Methods

#### 2.2.1. Solubility Studies

Aqueous solutions of SBE or HP (1%, 5%, 10%, 15%, 20% *w/w*) were initially prepared using HPLC-grade water. An excess amount of DDC-Cu was added separately to 1 mL of the aforementioned solutions. The resulting mixtures were vortexed, sonicated at room temperature for 2 h, then agitated at 250 rpm for 3 days at room temperature using Stuart reciprocating shaker SSL2 (Cole-Parmer, Neots, UK). Subsequently, the mixtures were centrifuged (Thermo Scientific Heraeus PIC017) for 10 min at 13,000 rpm. The supernatants (0.5 mL) were transferred to Spin X centrifuge tube with cellulose acetate filter (0.45 µm) and centrifuged again for 10 min at 13,000 rpm. The filtrates were further diluted with HPLC-grade water, and UV absorbance was measured at 435 nm using a UV spectrophotometer (Evolution TM 220 series with INSIGHT TM software, Thermo Fisher Scientific, Loughborough, UK). The DDC-Cu concentration was finally calculated based on a linear equation from a standard calibration curve of DDC-Cu in DMSO. The LOD and LOQ for DDC-Cu, which were predicted from the calibration curve equation, were 0.06 mg/L and 0.188 mg/L, respectively. The %RSD of the repeatability test at 100 mg/L drug concentration was less than 1.0, whereas the %RSD of the ruggedness test for the same drug concentration is less than 2.0. Full validation of the analytical methods is added as Appendix A. In addition, all results were confirmed using HPLC analysis according to our method described previously [1]. In Brief, UltiMate 3000 UHPLC (Thermo Fisher Scientific, Loughborough, UK) with Accucore C18 4.6 × 150 mm column with a 2.6 μm particle size (Thermo Fisher Scientific, Loughborough, UK) were used. The mobile phase comprised 90% HPLC-grade methanol and 10% HPLC-grade water. The flow rate was 0.5 mL/min, and UV detection was performed at 275 and 435 nm with an injection volume of 20 μL.

#### 2.2.2. Freeze-Drying

Samples (10 mL) of the above DDC-Cu cyclodextrin solutions (15% and 20% *w/v*) were frozen at −80 °C for 12 h then freeze-dried under negative pressure of 0.06 ± 0.01 using Lyotrap LTE freeze dryer (LTE Scientific Ltd, Oldham, UK) for 72 h. The samples (Table 1) were stored at room temperature for further studies.

#### 2.2.3. Differential Scanning Calorimetry (DSC) Analysis

DSC analysis of DDC-Cu, SBE, HP and freeze-dried formulations was performed using a differential scanning calorimeter (214 Polyma, Netzsch, Selb, Germany). Samples were weighted and crimp-sealed in aluminium led pierced-pans. Tests were carried out under nitrogen gas flow (20 mL/min) over a temperature range of 0–300 °C and at a heating rate of 10 °C/min. The thermograms were analysed using Proteus Analysis software (version 7.0.1).

#### 2.2.4. Thermogravimetric Analysis (TGA)

TGA was carried out using TG 209 F3 Tarsus model (Netzsch, Selb, Germany) in order to determine the moisture content of freeze-dried samples. Samples (3–7 mg) were loaded on an open fused silica crucible suspended from a microbalance and heated dynamically from 25 to 110 °C at 10 °C/min heating rate. The samples were kept at 110 °C for 3 min then heated to 300 °C at 10 °C/min. The resulting data were analysed using Proteus Analysis software (version 6.1.0, Netzsch, Selb, Germany).

#### 2.2.5. Fourier Transform Infrared Spectroscopy (FTIR)

Infrared spectra were obtained using Spectrum Nicolet iSF FTIR Spectrophotometer (Thermo Scientific, Loughborough, UK), supported with iDF ATR. Small amount of each sample was directly loaded into the instrument without any treatment. The analysed spectra showed the % transmittance of different samples versus the wavenumbers range of 4000 to 650 cm^−1^. The data were obtained by OMNIC 9 software and presented by Excel.

#### 2.2.6. Drug Solution Stability Study

The stability of freshly prepared and reconstituted solutions listed on Table 1 was assessed over 28 days. For reconstituted solutions, the freeze-dried samples were redissolved in HPLC-grade water without any further shaking or stirring. Samples were stored at room temperature and at appropriate intervals (weekly), aliquots of 0.5 mL were filtered and tested by UV spectrophotometer as explained above. The stability of DDC-Cu in solutions were confirmed by HPLC analysis as described earlier.

#### 2.2.7. Cytotoxicity Studies

The cytotoxic effect of SBE20 and HP20 freeze-dried formulations on breast cancer cells was assessed in vitro. Living cells, indicated by metabolic activity, were estimated using an MTT assay and presented as a percentage of the non-treated cells [19]. MDA-MB-231 and MDA-MB-231_PAC10_ cells were seeded in 96-well plates at a density of 10 × 10^3^ cells for MDA-MB-231 and 5 × 10^3^ cells for MDA-MB-231_PAC10_ in Dulbecco’s modified Eagle’s medium (DMEM) with 10% FBS, 1 mM sodium pyruvate, 2 mM L-glutamine, 50 IU/mL penicillin, 50 μg/mL streptomycin and 0.1 mM non-essential amino acids. Cells were incubated for 24 h (Heracell 150i CO2 incubator, Thermo Scientific, Loughborough, UK) at 37 °C with 5% CO_2_, then treated with full growth media containing reconstituted formulations. Blank CDs solutions were used as negative controls and DDC-Cu was used as a positive control. After 72 h, all wells were treated with a standard 3-(4,5-Dimethylthiazol-2-yl)-2,5-Diphenyltetrazolium bromide (MTT) assay as previously described [20]. The experiments were carried out in triplicate and the IC_50_ values were calculated.

#### 2.2.8. Statistical Analysis

Statistical significance was measured using the one-way analysis of variance (ANOVA) and student’s t-tests as appropriate. All values were expressed as the mean ± standard deviation. Values of *p* < 0.05 were regarded as significantly different.

## 3. Results and Discussion

### 3.1. Solubility Studies

The intrinsic solubility of DDC-Cu was below the limit of detection of both UV spectrometry (Appendix A) and even HPLC methods (data not shown). Thus, the intrinsic solubility was predicted in water using DMSO as a co-solvent by adopting methods of prediction reported by Vithlani et al. [21]. Our results confirmed that DDC-Cu was practically insoluble in water as the predicted intrinsic solubility was 0.0007 mg/L (Appendix A).

The solubility of DDC-Cu in different concentrations of HP and SBE was explored; the higher the CD concentration, the higher the apparent solubility of DDC-Cu (Figure 2). The solubility of DDC-Cu had a polynomial relationship with the increasing concentrations of both types of CD.

According to Higuchi and Connors’ model, this type of relationship between CDs and DDC-Cu that deviates positively from linearity was Ap-type phase (Figure 2B) [22]. Ap-type phase was usually obtained when water soluble CDs, such as HP and SBE, were used, and indicated that drug solubility increased in a non-linear fashion with increasing CDs concentration. This positive deviation suggests that the employed CDs were proportionately more effective at higher concentrations [12]. This might also be explained by the superior drug–CD interaction compared to CD–CD interactions. Normally, the physicochemical properties of the guest molecule and the type of CD (the host) predominate the outcome of this competition between both interactions [23]. The prepared solutions were clear with different degree of yellow, but become dark brown or black at higher concentrations, hence, 50-fold dilutions were used for imaging (Figure 2C).

It is worth mentioning that the pH values for all CD solutions remained stable within the range of 5–7 throughout the study (Appendix A). This might suggest the suitability of CD formulations for clinical applications.

As shown in Figure 2A, for both 20% *w/w* CD solutions, the solubility of DDC-Cu has reached approximately 4 mg /mL. This provides not only unprecedented concentration of DDC-Cu but also potentially stable freeze-dried formulations and stable solutions (tested for 28 days as shown below). Furthermore, this solubility value is appropriate for further investigation on anticancer activity of this ingredient. For example, the minimum limit of solubility to initiate pharmacokinetic studies on rats or mice is 2.0 mg/mL [24].

### 3.2. Physicochemical Characterisations Studies of Freeze-Dried Samples

#### 3.2.1. Differential Scanning Calorimetry (DSC) Analysis

The thermograms of SBE, HP and DDC-Cu raw materials and SBE15, SBE20, HP15 and HP20 freeze dried formulations are shown in Figure 3. Free DDC-Cu had a sharp endothermic peak at 201 °C attributed to a melting temperature with relatively high enthalpy (ΔH) value (−80.65 J/g). Melting temperature and heat of fusion (enthalpy) of a material might be used to indicate crystallinity; the higher the crystallinity, the higher the melting temperature. Therefore, any drop in the melting point of a “guest” molecule might indicate a stronger interaction with the “host” (the macromolecule) [25,26]. Correspondingly, all freeze-dried formulations for both HP and SBE showed no endothermic peaks in the region of that shown by the free DDC-Cu crystals. This might be an indicator of the solid solution state of DDC-Cu included in the freeze-dried formulations. Experimentally, this could be easily noticed by the instant dissolution of the formulations in water producing solutions with development of a distinctive yellow colour at low concentrations (Figure 2C), and a very dark green colour at higher concentrations (not shown). The broad peaks around 100 °C shown by thermograms of freeze-dried formulations might be explained by the evaporation of residual moisture in these formulations.

#### 3.2.2. Thermogravimetric Analysis (TGA)

The thermal stability of the formulations compared to raw materials was further investigated using thermogravimetric analysis (TGA). Figure 4 shows that pure DDC-Cu experienced three stages of weight loss: early stage of 40 to 100 °C, the weight loss was due to loss of surface adsorbed and hydrogen bond water. This stage is less noticeable for CD raw materials and formulations. No further significant decrease in weight is shown during the second stage for DDC-Cu (100–190 °C) and CD raw materials and formulations (100–260 °C). Starting at approximately 190 °C, the third stage for DDC-Cu involves a sharp loss of the total weight at 290 °C. This might be explained by the cleavage of chemical bonds between DDC and Cu^2+^, followed by the degradation of reactive DDC to diethylamine and carbon disulfide [26]. The third stage starts at two different points for CD formulations; at approximately 260 °C SBE formulations, while HP formulations are expected to start above 300 °C [27]. The difference between TGA profiles for both beta CDs, owning to different chemical structures, have already been reported in literature [27,28,29]. More importantly, SBE15 and SBE20 formulations showed more resistance to degradation than raw SBE, by around 15 °C between both points where a sharp drop in weight occurred (Figure 4A). This may be due to the fact that the interaction between DDC-Cu and SBE resulted in stronger complex formation, demanding more heat to break. Interestingly, the distinctive degradation peak of DDC-Cu is not shown by both CDs formulations, which might constitute an evidence of the protection provided by the CDs as a stabilising host for DDC-Cu [30].

#### 3.2.3. Fourier Transform Infrared Spectroscopy (FTIR)

Figure 5 shows the FTIR spectra of the freeze-dried formulations of SBE and HP, pure DDC-Cu and pure CDs. The IR spectrum of DDC-Cu shows a strong absorption band at 1504 cm^−1^ corresponding to the C-N bond, which indicates its partial double bond nature. There was no difference between pure CDs spectra and the corresponding freeze-dried formulations, where none of DDC-Cu vibration bands stands significantly in the CD complexes. The disappearance of DDC-Cu vibration bands in the formulations might mainly be due to the fact that the formulation content of DDC-Cu was lower than the detection limit of our FT-IR kit. However, the restriction of bending and stretching vibration of guest molecules by the formation of inclusion complexes have also been reported [31].

### 3.3. Drug Solution Stability Studies

The rapid precipitation of DDC-Cu in the form of crystals is a main challenge during the preparation of any formulation incorporating this material. In addition, most of the previous attempts have failed to produce stable solutions of DDC-Cu that are potentially suitable for therapeutic administration. The stability of freshly prepared and reconstituted solutions was performed over 28 days to confirm that there was no precipitation, crystallisation or degradation of DDC-Cu. As shown by Figure 6, all solutions were stable for 28 days without any significant change (*p* > 0.05) on DDC-Cu concentration. Compared to other nanomedical formulations [10,32], no sign of precipitation nor crystallisation was shown. These results were not unexpected for CDs used successfully in pharmaceutical research as solubility enhancers of hydrophobic drugs/ingredients [33,34].

### 3.4. Cytotoxicity Studies

Triple-negative breast cancer (TNBC) is an aggressive type of cancer due to the lack of molecular markers such as oestrogen receptors and progesterone receptors [35]. Furthermore, TNBC can develop chemoresistance after repetitive exposure to first line chemotherapy agents such as Paclitaxel (PAC), an anti-cancer drug [5,36,37,38]. This acquired chemoresistance then becomes the major challenge in chemotherapy. In this study, two types of TNBC cell lines: (1) MDA-MB-231, the sensitive and (2) MDA-MB-231_PAC10_, the resistant cell line were used to assess the cytotoxic effect of DDC-Cu and its formulations. In addition to its high resistance to PAC, MDA-MB-231_PAC10_ has been reported to be cross-resistant to drugs such as cisplatin, docetaxel and doxorubicin [5].

As shown in Figure 7, MDA-MB-231 cells were sensitive to the cytotoxic effect of PAC and DDC-Cu. However, MDA-MB-231_PAC10_ showed high resistant to PAC (IC_50_ >1000 nM) but completely terminated by DDC-Cu (*p* < 0.05) (Figure 7C). These results confirm that DDC-Cu had the ability to overcome chemoresistance of MDA-MB-231_PAC10_. Interestingly, Liu et al. (2013) have reported similar results but after applying disulfiram in combination with CuCl_2_ on both MDA-MB-231_PAC10_ [5]. This also supports our previous findings showing that the mechanism of action for the combined disulfiram and CuCl_2_ may not be limited to ROS and that DDC-Cu has a strong and prolonged anti-cancer activity (i.e., potential druggable anti-cancer candidate) [1].

The cytotoxic effect of inclusion complexes of DDC-Cu (HP20 and SBE20) was also assessed. As shown in Figure 8, HP20 and SBE20 elicited cytotoxic activity similar to that of free DDC-Cu, used as a positive control, on both cell lines. Furthermore, there were no significant differences amongst HP20 and SBE20, and DDC-Cu (*p* > 0.05), whereas empty CDs solutions had no cytotoxic activity on both cells. Despite the apparent differences in IC_50_ values between both CDs HP20 and SBE20, and both cell lines MDA-MB-231, and MDA-MB-231_PAC10_, our statistical analysis confirmed that there was no significant difference to report (*p* > 0.05) (Figure 8).

## 4. Conclusions

The use DDC-Cu as an anticancer drug has always been challenged by its poor aqueous solubility. In this work, we have shown that the solubility of DDC-Cu may be improved using CD inclusion complexes. The freeze-dried formulations were readily dissolved in water yielding DDC-Cu solutions of high stability for 28 days. The thermal characteristics of the freeze-dried formulations supported the formation of inclusion complexes with the DDC-Cu molecules interacting CD molecules in a solid solution form. The MTT cytotoxicity study showed a strong toxic effect of the freeze-dried formulations on both MDA-MB-231 (sensitive) and MDA-MB-231_PAC10_ (chemoresistant) breast cancer cell lines similar to that of free DDC-Cu. Our results report that cyclodextrin DDC-Cu complexes have a great potential for further anticancer applications, especially, in vivo studies.

## Figures and Tables

**Figure 1 pharmaceutics-13-00084-f001:**
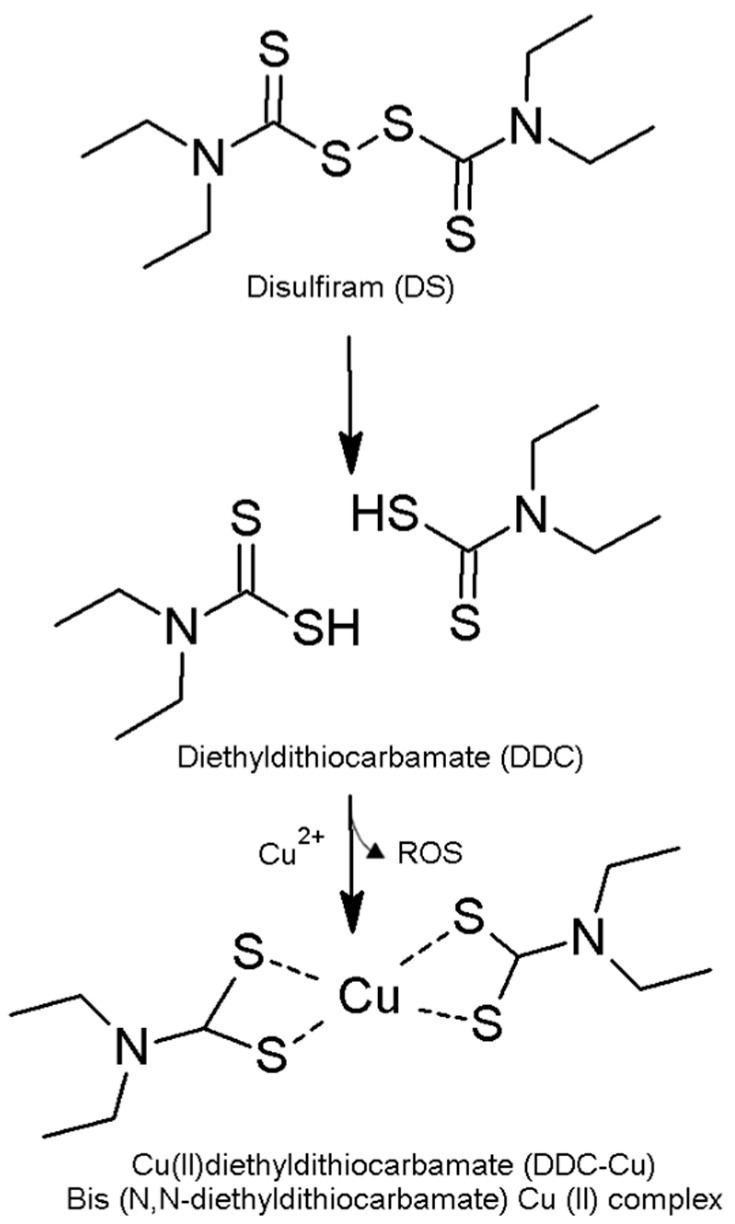
The formation of diethyldithiocarbamate copper II (DDC-Cu) as a result of disulfiram (DS) reaction with copper [3].

**Figure 2 pharmaceutics-13-00084-f002:**
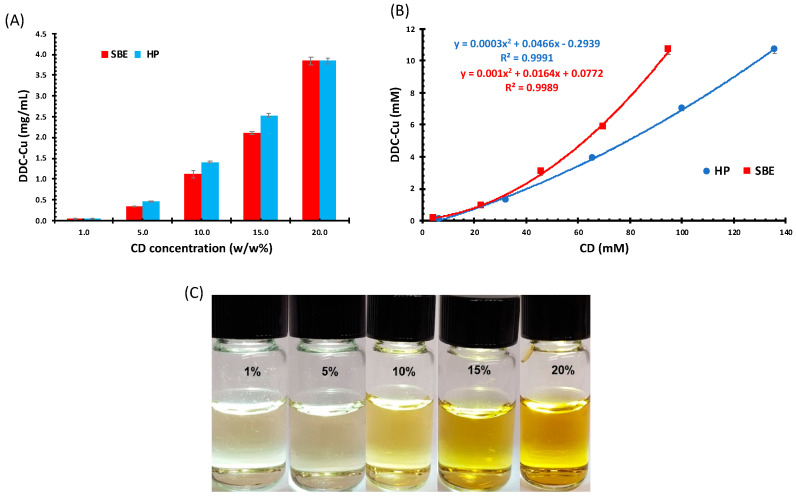
(**A**) Diethyldithiocarbamic acid (DDC)-Cu Solubility (mg/mL) in assorted CD solutions (mM), (**B**) Phase solubility diagrams of DDC-Cu/CD (mM) (Mean ± SD; *n* = 3), and (**C**) DDC-Cu saturated solutions in serial concentrations of sulfobutyl ether beta-cyclodextrin (SBE), (50-fold dilutions).

**Figure 3 pharmaceutics-13-00084-f003:**
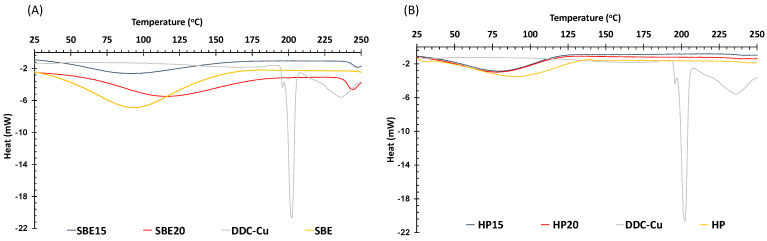
DSC thermographs of (**A**) SBE-CD freeze dried formulations, SBE-CD and DDC-Cu and (**B**) HP-CD freeze dried formulations, HP-CD and DDC-Cu.

**Figure 4 pharmaceutics-13-00084-f004:**
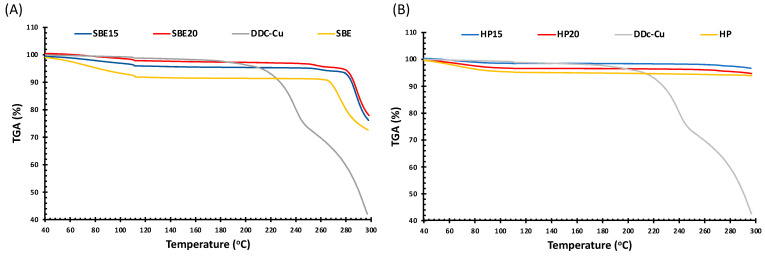
Thermogravimetric Analysis (TGA) thermographs of (**A**) SBE-CD freeze dried formulations, SBE-CD and DDC-Cu and (**B**) HP-CD freeze dried formulations, HP-CD and DDC-Cu.

**Figure 5 pharmaceutics-13-00084-f005:**
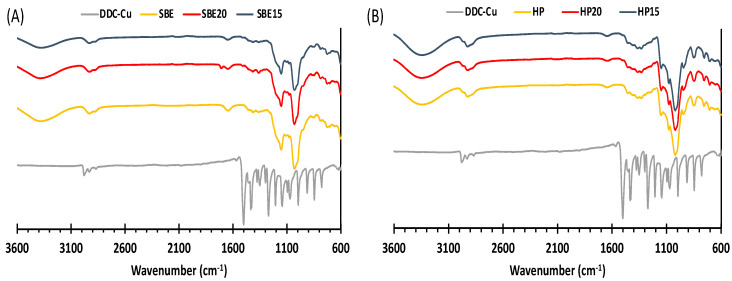
FTIR spectra of (**A**) SBE freeze dried formulations, SBE and DDC-Cu and (**B**) HP freeze dried formulations, HP and DDC-Cu.

**Figure 6 pharmaceutics-13-00084-f006:**
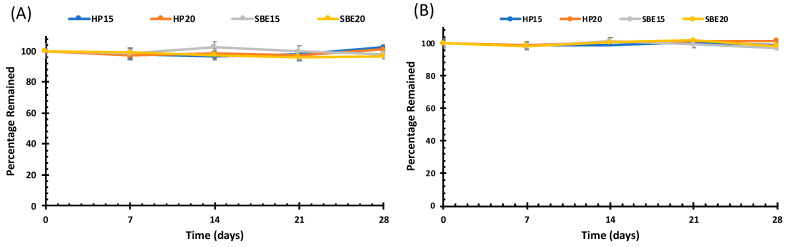
Solution state stability studies of the freshly prepared (**A**) and reconstituted (**B**) DDC-Cu CD solutions. For both, the axes y represents the percentage remained of solubilized DDC-Cu in solution.

**Figure 7 pharmaceutics-13-00084-f007:**
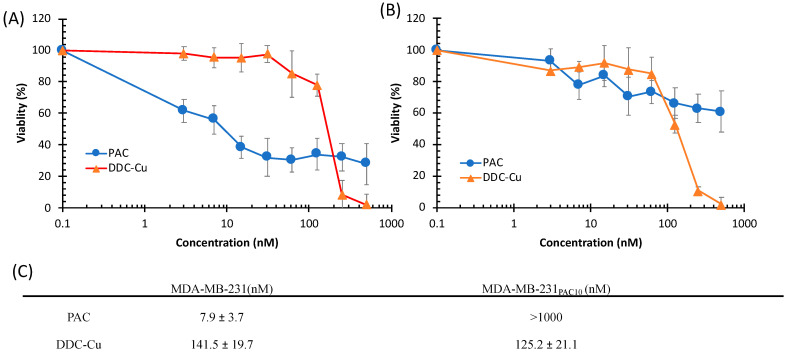
Survival curves (MTT assay) of triple negative breast cancer cell lines MDA-MB-231 (**A**) and MDA-MB-231_PAC10_ (**B**) with increasing concentrations of paclitaxel (PAC) and DDC-Cu (*n* = 3 ± SD). (**C**) The IC_50_ values of PAC and DDC-Cu on both breast cancer cell lines (*n* = 3 ± SD).

**Figure 8 pharmaceutics-13-00084-f008:**
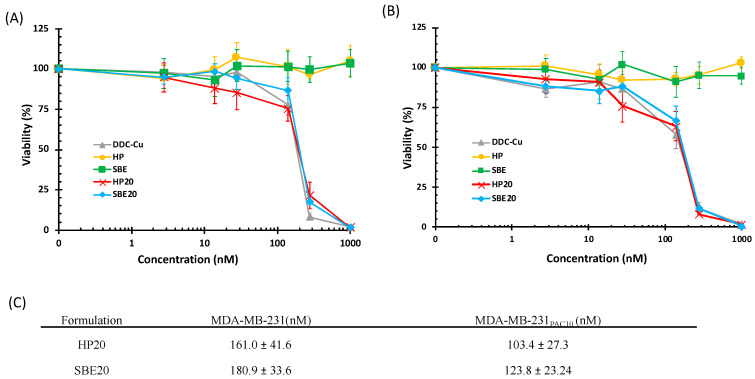
Survival curves (MTT assay) of triple negative breast cancer cell lines MDA-MB-231 (**A**) and MDA-MB-231_PAC10_ (**B**) after 72 h of exposure to increasing concentration of DDC-Cu formulations. (**C**) The IC_50_ values of DDC-Cu and formulations on both breast cancer cell lines (*n* = 3 ± SD).

**Table 1 pharmaceutics-13-00084-t001:** Cyclodextrin (CD) concentration and acronyms of the freeze-dried formulations.

Sample Name	CD Type	CD Concentration (% *w/w*)	DDC-Cu Concentration mg/mL (Mean ± SD; *n* = 3)
SBE15	SBE	15	2.11± 0.03
SBE20	SBE	20	3.85± 0.09
HP15	HP	15	2.51 ± 0.06
HP20	HP	20	3.85 ± 0.07

## Data Availability

The data presented in this study are complemented by Appendix A. Enquiries may be made to the corresponding author.

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
