# Peer review of "Cyclodextrin Diethyldithiocarbamate Copper II Inclusion Complexes: A Promising Chemotherapeutic Delivery System against Chemoresistant Triple Negative Breast Cancer Cell Lines"

_pharmaceutics, 2021, doi:10.3390/pharmaceutics13010084_

Round 1

Reviewer 1 Report

  1. I didn’t understand the logic in lines 49- 54 – whether disulfiram has anti-cancer effects that are wholly copper dependant on are in part copper dependant and why there may be pathways other than ROS that contribute to the cytotoxicity. Please reword these few sentences to make it clearer.
  2. Is there any evidence that DDC-Cu preferentially kills cancer cells compared to normal cells? This could usefully be presented in the introduction
  3. Line 64. Is bioavailable the right term? Do the authors mean tumour uptake?
  4. Lines 64-69 minor grammar improvements would help (use the appropriate tense). Some minor grammar improvements are desirable throughout the manuscript.
  5. Some abbreviations are not defined when first used. Eg line 76
  6. Line 155 and elsewhere in the manuscript: MTT does not measure cytotoxicity, it measures metabolic activity which is a surrogate for relative cell number
  7. Lines 174-179 belong in the introduction, not the results section.
  8. Figure 4 - panel A is not referenced in the legends
  9. Figure 7 the x axis needs to be relabelled (conc is inappropriate) and in general the authors should avoid the use of unneeded abbreviations (eg PAC). MTT does not measure viability (y axis in fig 7 and 8)
  10. Lines 290-300 there is no need to include IC50 values in the text because they are already in the table.
  11. The use of the term “pan” on line 296 is not justified
  12. I see no evidence to support the authors assertions about mechanism of action on lines 299-300
  13. Figure 9 – these images are not of adequate quality to assess “apoptotic features” and should be replaced or deleted

Reviewer 2 Report

The article entitled "Cyclodextrin Diethyldithiocarbamate Copper II Inclusion Complexes: A Promising Chemotherapeutic Delivery System Against Chemoresistant Triple Negative Breast Cancer Cell Lines"  and written by Ammar Said Suliman and co-workers reports on a interesting anticancer activity study against Triple negative Breast Cancers Cells  by using complexes of DDC-Cu with hydroxypropyl beta-cyclodextrin (HP) or sulfobutyl ether beta-cyclodextrin (SBE). The manuscript is sound and well strucured. Results are coherently discussed but it requires few amendments\corrections prior acceptance:

I have listed following issues:

1) A moderate English revision by native speaker is required in order to remove some grammatical typos or language impefections

2) A "targeted" literature update by adding\updating recent references\reviews is warmly suggested

3) A thorough CLSM\FACS Complexes co-localisation\uptake pathways analysis\discussion is neglected: in my opinion it should be either performed or otherwise justified\discussed

4) What was the rationale for the choice of cells? Any difference by using different type of cancer? Please comment

5) What are the prospectives for Animal models\in vivo study? Please outline a future plan in this direction.

6) Going further: are there already direct patients \medical practical applications of this type of complexes? Discuss this point.
